# Nutrient Composition of Demersal, Pelagic, and Mesopelagic Fish Species Sampled Off the Coast of Bangladesh and Their Potential Contribution to Food and Nutrition Security—The EAF-Nansen Programme

**DOI:** 10.3390/foods9060730

**Published:** 2020-06-03

**Authors:** Anna Nordhagen, Abu Ansar Md. Rizwan, Inger Aakre, Amalie Moxness Reksten, Lauren Michelle Pincus, Annbjørg Bøkevoll, Al Mamun, Shakuntala Haraksingh Thilsted, Thaung Htut, Thiruchenduran Somasundaram, Marian Kjellevold

**Affiliations:** 1Institute of Marine Research, P.O. Box 2029 Nordnes, 5817 Bergen, Norway; nordhagen_94@hotmail.com (A.N.); inger.aakre@hi.no (I.A.); annbjorg.bokevoll@hi.no (A.B.); marian.kjellevold@hi.no (M.K.); 2Health and Nutrition, Social Assistance and Rehabilitation for the Physically Vulnerable (SARPV), Cox’s Bazar 4700, Bangladesh; aamdrizwan@gmail.com; 3WorldFish, Jalan Batu Maung, Batu Maung, Bayan Lepas, Penang 11960, Malaysia; l.pincus@cgiar.org (L.M.P.); s.thilsted@cgiar.org (S.H.T.); 4Marine Fisheries Survey Management Unit, Department of Fisheries, CGO Building-2, Agrabad, Chattogram 4100, Bangladesh; mamunbau08@yahoo.com; 5Wildlife Conservation Society-Myanmar Program, P.O. Box Kamayut, Yangon 11041, Myanmar; thtut@wcs.org; 6Institute of Postharvest Technology, National Aquatic Resources Research and Development Agency (NARA), P.O. Box Colombo 01500, Sri Lanka; tsomasundaram@deakin.edu.au

**Keywords:** fish, nutrient composition, recommended nutrient intakes, micronutrients, food and nutrition security, Bangladesh, Bay of Bengal, marine, food composition data, mesopelagic

## Abstract

Fish is a major part of the Bangladeshi diet, but data on the nutrient composition of marine fish species are sparse. Mesopelagic fish may be a new potential resource of food and nutrients; however, nutrient composition data are lacking. The aim of this study was to provide nutrient composition data of fish species sampled off the coast of Bangladesh and determine their potential contribution to recommended nutrient intakes (RNI). Seven species from the pelagic, mesopelagic, and demersal zones were sampled from the coast of Bangladesh with Dr. Fridtjof Nansen in 2018. Three pooled samples containing 15-840 individuals from each species were analysed at the Institute of Marine Research, Norway. The demersal species contained substantially lower concentrations of nearly all nutrients, whereas the mesopelagic species generally were more nutrient dense. All species, except for the demersal species Bombay duck (9% dry matter), were found to contribute ≥100% to the RNI of vitamin B_12_, eicosapentaenoic acid, docosahexaenoic acid, and selenium. All species, except for the demersal fish species, contributed ≥25% to the RNI of six or more nutrients. The data presented in this paper are an important contribution to the Bangladeshi food composition table and contribute to the understanding of fish as an important source of micronutrients.

## 1. Introduction

Seafood is considered a high protein food and a major source of fatty acids, such as the very long-chain omega-3 polyunsaturated fatty acids eicosapentaenoic acid (EPA) and docosahexaenoic acid (DHA). Fish is also a rich dietary source of several micronutrients, such as vitamin A, vitamin B_12_, vitamin D, calcium, iodine, selenium, and zinc [1,2,3]. Both freshwater and marine fish are popular among the Bangladeshi people; fish is readily available, well-liked, and less expensive than other animal-source foods in Bangladesh [4,5]. Fish is the third most commonly consumed food in the Bangladeshi diet after rice and vegetables [4] and is by far the most important source of animal protein, accounting for approximately 60% of total animal protein intake per capita per year [5,6,7,8]. In the past two decades, the consumption of fish in Bangladesh has increased because of the rapid expansion of aquaculture [5,9,10]. However, malnutrition remains widespread, with a 36% prevalence rate of stunting and a 14% prevalence rate of wasting in children under 5 years of age—both of which are among the highest globally [11,12]. Undernutrition is exacerbated by poor dietary diversity and micronutrient deficiencies continue to be a major problem in the country [5,10,13]. Vitamin A, vitamin B_12_, zinc, iodine, and folate deficiencies, as well as maternal and child anaemia, are particular public health issues and are highly prevalent [9]. 

Food composition data (FCD) represent a basic and essential tool for planning interventions to improve health, nutrition, and food and nutrition security in all populations [14]. FCD are used in all aspects of nutrition: food labeling and diet formulation, assessment of nutrient intake and requirements, education and research, estimations of food and nutrition security prevalence, and public health policy formulation [14,15]. The first food composition table (FCT) for Bangladesh was developed in 1977 and consisted of a combination of borrowed and analytical data. The FCT lacked proper documentation of the origin of the data and the analytical methods utilised. It also contained a large number of missing nutrient values and lacked precise descriptions of the foods. In 2013, as a response to the long-standing demands from several institutions in the health, nutrition, education, and agriculture sectors, Bangladesh developed the new and improved FCT in use today. This FCT contains 381 foods and 18 nutrients, with most of the data given for raw foods. The foods are classified into 15 food groups, with “Fish, shellfish, and their products” being a major group consisting of 72 food items. Only nine marine fish species are included in the FCT, whereas around 511 various marine fish species exist in the marine waters of Bangladesh [16]. As stated in the FCT, there is still a major lack of knowledge on the nutrient composition of fish species available for consumption in Bangladesh [17]. 

Though Bangladesh has huge marine water resources, it is estimated that approximately 15% of the country’s total fish production stems from marine capture fisheries, with aquaculture presently providing more than half of the fish available for direct human consumption [8,18]. According to official statistics, the most commonly consumed fish species are all of freshwater origin and includes species such as tilapia, catfish, and mrigal carp. However, these data may fail to include a major part of the total fish production, which includes small indigenous fish species, often thought of as “weed” fish [19,20]. The most commonly consumed marine species are hilsa shad and other shads [21], closely followed by Bombay duck [22]. In the 1970s and 1980s, several surveys examined the status of the marine fish stocks in the country, suggesting that these might be over-exploited, but no recent or comprehensive knowledge is available [18]. To reduce existing undernutrition and micronutrient deficiencies and contribute to food and nutrition security, new marine resources could be explored, not only directly for human consumption, but also indirectly as feed ingredients for use in the flourishing aquaculture sector. Mesopelagic fish, the smallest and most abundant pelagic fish species residing in the mesopelagic zone stretching from 200 to 1000 m depth during daytime, may represent an important and sustainable source of marine nutrients, however, present research and knowledge on these species are particularly scarce [23,24,25]. The aim of this study was to quantify the nutritional composition of pelagic, mesopelagic, and demersal fish species from the marine waters of Bangladesh to fill knowledge gaps in the existing FCD. In addition, the potential contribution of these fish species to recommended nutrient intakes as a mean to evaluate their value to food and nutrition security in Bangladesh was assessed. The micronutrients considered in relation to the RNI were vitamin A, vitamin B_12_, vitamin D, calcium, iron, iodine, selenium, and zinc, and the essential fatty acids EPA and DHA. 

## 2. Materials and Methods 

This paper uses data collected through the scientific surveys with the research vessel (R/V) Dr. Fridtjof Nansen as part of the collaboration between the EAF-Nansen Programme and the Department of Fisheries (DoF) in Bangladesh. The EAF-Nansen Programme is a partnership between the Food and Agriculture Organization of the United Nations (FAO), the Norwegian Agency for Development Cooperation (Norad), and the Institute of Marine Research (IMR), Norway, for sustainable management of the fisheries of partner countries. 

### 2.1. Fish Sampling

Sampling of fish was performed during a survey conducted by R/V Dr. Fridtjof Nansen in the Bay of Bengal off the coast of Bangladesh. Surveying took place over twelve days in August 2018. Sampling of fish was carried out using pelagic (MultiPelt 624) and bottom trawls (Gisund Super bottom trawl). “Pelagic” refers to the pelagic zone of the ocean, and fish in this habitat zone occupy the open water column that is not near the shore nor the bottom, whereas demersal fish live on or near the bottom. The sampling location for each species is presented in a map in Figure 1. For each trawl, the fish were sorted, and the species identified by local taxonomists using the FAO Species Guides and catalogues [26,27,28]. A list of prioritised fish species based on the most commonly consumed species in Bangladesh was generated at the beginning of the survey by Bangladeshi scientists and taxonomists. Mesopelagic species were included in the study due to the limited knowledge on the nutrient composition of these species, even though they are currently not widely consumed. Samples of pelagic and demersal fish species were prepared according to local dietary habits: as fillets with skin and bones, but without the head, tail, and viscera (Table 1). The mesopelagic species were prepared whole, including the head, skin, tail, and viscera. Three pooled samples, consisting of a minimum of five individual fish in each sample, were prepared for nutrient analyses at the IMR in Bergen, Norway. The samples were homogenised using a food processor (Braun Multiquick 7 K3000, Kronberg im Taunus, Germany) and stored as wet samples at −20 °C in the on-board freezer. After a minimum of 12 h in the freezer, a sub-sample of each wet sample was freeze-dried for 72 h (24 h at −50 °C, immediately followed by 48 h at +25 °C, with a vacuum of 0.2–0.01 mbar), using a Labconco FreeZone 18 litres freeze-dryer (mod. 7750306, Kansas City, MO, USA) and the dry matter (%) was determined. Freeze-dried samples were then homogenised once more using a knife mill (Retch Grindomix GM 200, Haan, Germany). All samples were packed in plastic bags and stored in insulated boxes in the vessel’s freezer (−20 °C) until shipment by air cargo to the IMR laboratories in Bergen, Norway, where the samples were stored at −80 °C, pending analyses. A detailed description of the sampling procedures can be found in Reksten et al. [29].

### 2.2. Analytical Methods

The determination of nutrients in the sampled species was carried out at the IMR laboratories. Details of the analytical methods are described by Reksten et al. [29]. Crude fat was extracted with ethyl acetate and filtered before the solvent was evaporated and the fat residue was weighed. The method is standardised as a Norwegian Standard, NS 9402 [30]. Crude protein was determined by burning the material in pure oxygen gas in a combustion tube at 950 °C. Nitrogen (N) was detected with a thermal conductively detector and the content of nitrogen was calculated from an estimated average of 16% N per 100 g protein. The following formula was used: N g/100 g × 6.25 = g protein/100 g, in accordance with the method accredited by the AOAC Official Methods of Analysis [31]. For analysis of fatty acids, lipids from the samples were extracted according to Folch et al. [32]. After filtering, the remaining samples were saponified and methylated. The fatty acid composition of total lipids was analysed as previously described [33,34]. The samples for the determination of vitamin A_1_ (sum of all trans-retinol and 13-, 11-, 9 cis retinol) and vitamin A_2_ (3,4 didehydro-all-trans-retinol) were saponified and the unsaponifiable material was extracted. Vitamin A_1_ and A_2_ concentrations were determined by high-performance liquid chromatography (HPLC, normal phase) using a Photo Diode Array (PDA) detector. The content of all-trans-retinol was calculated by external calibration (standard curve) [35], and the content of the other vitamin A forms was calculated based on the external calibration curve for all-trans-retinol multiplied by a correction factor. The samples for the determination of vitamin D_3_ were saponified and the unsaponifiable material was extracted and then purified on a preparative HPLC column. The fraction containing D_2_ (ergocalciferol) and D_3_ (cholecalciferol) was pooled (normal phase). This fraction was injected on an analytical HPLC column (reverse phase). Vitamin D_2_/D_3_ was determined by an UV detector. The content of vitamin D_3_ was calculated using an internal standard (vitamin D_2_) [36]. Vitamin B_12_ (cobalamin) was released from the sample by extraction (autoclaving in acetate buffer) and mixed with growth medium, added a microorganism (*Lactobacillus delbruecki* -ATCC 4797), and incubated at 37 °C for 22 h. The vitamin B_12_ content was calculated by comparing the growth of the organism in the unknown samples, with the growth of the organism in known standard concentrations with turbidimetric reading (Optical Density, OD, v/575 nm) [37]. The concentrations of minerals (selenium (Se), zinc (Zn), iron (Fe), calcium (Ca), potassium (K), magnesium (Mg), phosphorus (P), and sodium (Na)) were determined by inductively coupled plasma-mass spectrometry (iCapQ ICP-MS, ThermoFisher Scientific, Waltham, MA, USA) equipped with an auto-sampler (FAST SC-4Q DX, Elemental Scientific, Omaha, NE, USA) after wet digestion in a microwave oven (UltraWave, Milestone, Sorisole, Italy), as described by Julshamn et al. [38]. The concentration of these minerals was quantified using an external standard curve in addition to an internal standard [39]. Three slightly different methods were applied: (1) for Ca, Na, K, Mg, and P, using scandium (Sc) as the internal standard, (2) for Zn and Se, using rhodium (Rh) as the internal standard, and (3) for iodine (I), tellurium (Te) was used as the internal standard. For the determination of I, the sample preparation was a basic extraction with tetramethylammonium hydroxide (TMAH) before ICP-MS analysis. 

### 2.3. Data Management and Presentation of Analytical Data

All analytical values were exported from Laboratory Information Management System (LIMS) into Microsoft^®^ Office Excel 2013 version 15.0 for calculations of means and standard deviations (SD). For species that were sampled from multiple locations, a mean value for all samples of each species was calculated. Single values < Limit of Quantitation (LOQ) are presented as below the respective value for each analyte. For vitamin A_2_, four values were below the LOQ of 0.5 µg/100 g, whereas for vitamin D, five values were below the LOQ of 1.0 µg/100 g. When calculating means and SD, values below LOQ were entered into the dataset as the respective LOQ-value divided by two.

### 2.4. Calculation of Potential Contribution to Recommended Nutrient Intakes

The potential contribution of each species to recommended nutrient intakes (RNI) of selected micronutrients was calculated within reference to non-pregnant, non-lactating, healthy females of reproductive age (19–50 years). A 100 g portion of raw, edible parts of the various fish species was used to estimate the percentage of the RNI for the various micronutrients, using the vitamin and mineral requirements published by WHO and FAO [40]. For the fatty acids EPA and DHA, the Dietary Reference Values from the European Food Safety Authority (EFSA) were used [41]. For species that were sampled at several different locations, the mean of all samples was calculated and used as the reference value for that species. Although variable for different nutrients, the RNI values were calculated assuming a dietary bioavailability of 100%.

## 3. Results

### 3.1. Sample Characteristics

This paper includes a total of five commonly consumed marine fish species and two mesopelagic fish species. For two of the pelagic species, samples from three separate locations were analysed, and for one of the demersal species, samples from two separate locations were analysed. The weight, length, and habitat characteristics of the sampled species are described in Table 2. 

### 3.2. Pelagic Fish Species

Table 3 describes the proximate composition of the sampled fish species. The pelagic species fringescale sardinella, slender rainbow sardine, and torpedo scad had the highest protein content of the sampled species (21/100 g raw, edible parts). Furthermore, a sample of fringescale sardinella presented the highest content of zinc compared to the rest of the species (2.0 mg/100 g raw, edible parts, Table 4), whereas torpedo scad had the highest content of vitamin B_12_ (1.1 µg/100 g raw, edible parts, Table 5). 

### 3.3. Mesopelagic Fish Species

The two mesopelagic species sampled whole, spinycheek lanternfish and unicorn cod, presented higher values of calcium, iodine, selenium, and iron than the rest of the sampled species. The vitamin A_1_ content was also substantially higher in both mesopelagic species, with a maximum value of 280 µg/100 g raw, edible parts in unicorn cod, compared to the pelagic and demersal species. The content of selected fatty acids is presented in Table 6. Spinycheek lanternfish was the most significant source of DHA and had the second highest content of EPA. 

### 3.4. Demersal Fish Species

The content of dry matter, total protein, and fat was considerably lower in Bombay duck compared to the other sampled species. Bombay duck also contained the lowest content of all fatty acids and all minerals, with the exception of zinc. The two demersal species, Bombay duck and longfin mojarra, presented the lowest overall values for all vitamins compared to the rest of the species. For vitamin A_2_ and vitamin D, all values were <LOQ (<0.5 µg/100 g raw, edible parts and <1.0 µg/100 g raw, edible parts, respectively). 

### 3.5. Potential Contribution to Recommended Nutrient Intakes

The various species’ potential contributions to the RNI of women of reproductive age for several micronutrients are presented in Figure 2 and Figure 3. With a consumption portion of 100 g raw edible parts, all species, except for the two demersal species Bombay duck and longfin mojarra, have the potential to contribute ≥100% to the RNI of vitamin B_12_ for women of reproductive age. Similarly, for EPA and DHA, all species, except for Bombay duck, may contribute ≥100% to the RNI. For vitamin A, most species may only contribute ≤10% to the RNI, except for the two mesopelagic species spinycheek lanternfish and unicorn cod, which were identified to potentially contribute ≥25% and ≥50%, respectively. For vitamin D, all sampled species may potentially contribute ≥40% to the RNI, except for the two demersal species, which both presented values <LOQ. Furthermore, all species, except for Bombay duck, may contribute ≥100% to the RNI of selenium. The two mesopelagic species were both identified to contribute ≥100% to the RNI of iodine, whereas the remaining species may contribute <25% to the RNI of this mineral. Both mesopelagic species may also contribute ≥100% to the RNI of calcium, whereas the rest of the species were all <30% in calcium levels. Neither of the species were identified to have the potential to contribute >15% to the RNI of iron.

## 4. Discussion

This paper presents extensive analytical information on the nutrient composition of several marine fish species from the pelagic, mesopelagic, and demersal zones sampled off the coast of Bangladesh in the Bay of Bengal. Of the seven fish species sampled, five are readily available and commonly consumed pelagic and demersal species, whereas the two mesopelagic species represent a group of fish species which have yet to be evaluated for their potential impact on food and nutrition security. Nutrient content varied considerably between the species; with a clear pattern of the mesopelagic species containing the highest concentrations of all micronutrients and the demersal species, particularly Bombay duck, containing substantially lower concentrations of nearly all nutrients. However, it is important to consider that the analyses were performed on raw samples and the water content in this species was particularly high. Several species, particularly the mesopelagic species, were also identified to have the potential to contribute substantially to the RNI of women of reproductive age for micronutrients important for food and nutrition security in Bangladesh. A multitude of studies have previously been conducted to identify the nutrient composition of freshwater fish species, both non-farmed and farmed, in Bangladesh [4,42,43,44,45,46,47,48,49], however, very few studies have been conducted to quantify the nutrients in wild, marine fish species. This study is the first to include comprehensive analyses of the nutrient composition, including the vitamin and mineral composition, of the two mesopelagic species, spinycheek lanternfish and unicorn cod. 

### 4.1. Pelagic Fish Species

None of the sampled pelagic species in this paper are included in the Bangladeshi FCT [17]. Torpedo scad is, however, listed in the Malaysian FCT, with nutrient values in agreement with our results for protein, fat, and iron (20.4/100 g, 1.3/100 g, and 2.5 mg/100 g, respectively) [50]. However, the values for vitamin A_1_ and calcium are considerably lower (29 µg/100 g and 64.0 mg/100 g, respectively) in the Malaysian FCT compared to the values presented in this paper (11.7 µg/100 g and 670 mg/100 g raw, edible parts, respectively). This is probably due to the inclusion and exclusion of various anatomical fish parts in the analyses; only the fillet was included in the samples in the Malaysian FCT [51], whereas we included the fillet with skin and bones, as this is the way these fish species are commonly consumed by the Bangladeshi population. The presence of vitamin A_2_ in freshwater fish, particularly in the eyes and liver, is well known, however, not many studies have evaluated concentrations in marine fish species. For fringescale sardinella, vitamin A_2_ comprised 54-77% of the total vitamin A content, thus highlighting the importance of analysing both isomers of the vitamin, even for marine species. The high percentage of vitamin A_2_ reported in this paper is also in line with the values reported for small indigenous species from Bangladesh [52], but further studies are required to evaluate the occurrence and significance of vitamin A_2_ in marine species. Nearly all calcium and most of the phosphorus and zinc are found primarily in the skin and bones of fish [53,54,55], thus, the inclusion of these edible parts in the analyses increases the concentration of these micronutrients. This was also demonstrated in a study with Indian fish species, including fringescale sardinella. The content of iron reported was 2.25 mg/100 g, which aligns with the values presented in this paper (2.2 mg/100 g raw, edible parts), whereas the calcium content of 164.3 mg/100 g reported in their study is substantially lower than the results presented in this paper (532 mg/100 g raw, edible parts). This difference in calcium content is likely also due to the difference in which anatomical parts of the fish were included in the analyses; they included only the fillet, thus excluding the calcium-rich bones and skin of the fish [56]. The calcium in small fish species commonly consumed whole have been confirmed to be of high bioavailability [53], with a similar absorption rate to that of milk [57]. 

### 4.2. Mesopelagic Fish Species

Mesopelagic species are small (usually 2–15 cm in length), deteriorate quickly after harvest and are presently not considered directly suitable for human consumption due to their unknown content of undesirable substances such as wax esters [23,24]. However, it should be recognised that, if unsuitable for direct human consumption, mesopelagic species may still be acceptable as ingredients in fish feed [23]. With the expanding aquaculture industry comes the need for wild-harvesting of marine fatty acids—a demand that mesopelagic species have been suggested to supply [58]. Myctophids, or lanternfish (one of the species also included in this paper), are the most abundant group of mesopelagic fish in the world’s oceans, and have, in a study by El-Mowafi et al., previously been identified with a satisfactory level of nutrients and a low level of contaminants and wax esters, and are thus presumed to be qualified to be utilised as ingredients in fish feed [59]. Lanternfish have also in previous studies been recognised as a highly attractive source of raw material in fish feed due to their high content of fatty acids, particularly EPA and DHA [60,61]. The two mesopelagic fish species included in this paper presented the highest nutrient content of all sampled species for EPA and DHA, but also for vitamin A_1_, calcium, iodine, and iron, as in accordance with results from Alvheim et al. [62]. The high micronutrient content in these species may also be naturally attributed to the inclusion of various fish parts in the analyses, such as bones, skin, and viscera [63]. The eyes and liver of fish have previously been identified as particularly rich sources of vitamin A [4,64], with approximately 90% of the vitamin A located in the eyes and viscera of the small indigenous species, mola (*Amblypharyngodon mola*) [65]. The scientific literature on the biochemical composition of mesopelagic species and the distribution of mesopelagic diversity is particularly scarce [58]. From a food and nutrition security perspective, these types of studies are crucial to better understand the potential contribution such species may have on local food and nutrition security as a potentially harvestable resource for future commercial exploitation, either directly through human consumption or in a more indirect way as feed ingredients in aquaculture. 

### 4.3. Demersal Fish Species

Bombay duck supplies the second largest marine fish catch in Bangladesh, representing over 12% of total marine catch, and is considered a very affordable and easily available food, highly consumed by the poor [22,66]. In this paper, Bombay duck has the lowest nutrient concentrations of all the sampled species for protein, fat, all fatty acids, all vitamins, and all minerals. Other studies have reported a moisture content of approximately 87% for this species [43,67], which is considered very high [68], and in line with the mean dry matter of only 8.8% for the two samples of Bombay duck in this paper. This may explain the low nutritional quality in raw samples, as an inverse relationship between the water content and the content of fatty acids and protein exists; when the water content increases, the content of other nutrients decreases [69,70]. However, Bombay duck is commonly consumed both as fresh fish and as dried fish [22], and the chosen style may influence the nutrient yield immensely. Mohanty et al. presented a crude protein and fat content of 8.2% and 2.2%, respectively, in wet samples of Bombay duck [67], similar to the values of 12.13% for protein and 2.15% for fat, as reported by Zaman et al. for wet samples [43]. When analysing the proximate nutrient content of dried Bombay duck sampled from the markets in Bangladesh, Siddique et al. (2012) reported a protein content of 57.03% and a fat content of 7.48%, leading to the conclusion that due to increased nutrient-density per 100 g in dried Bombay duck compared to fresh Bombay duck, the consumption of dried Bombay duck is preferred [71]. 

### 4.4. Potential Contribution to Recommended Nutrient Intakes

The availability and thus the consumption of fish is dependent on socioeconomic status and prices, and fish is considered expensive compared to staple foods such as rice or maize [72,73]. Lower priced fish, such as small pelagic species and small indigenous species, both commonly consumed whole, are therefore preferentially purchased among poorer Bangladeshi households, thus playing an important role for food and nutrition security by diversifying the diets of the poor [45,74,75]. In Bangladesh, farmed fish is now cheaper and more readily available than marine fish due to expansions of aquaculture achieved over the last thirty years [73]. However, from a food and nutrition security perspective, it is important to consider not only the quantity of fish available in the country, but also how these fish are contributing to the nutritional needs of the population. Aquaculture systems produce only a limited variety of species, thus creating a shift away from the consumption of a diversity of species with a wide array of nutritional profiles that follows the diverse capture fisheries. Despite the rapid growth of aquaculture and increased overall consumption of fish in recent years, the overall lower nutritional quality of farmed fish compared to marine fish have likely exacerbated existing micronutrient deficiencies in Bangladesh [10,42]. With the exception of Bombay duck, we found that all marine species sampled may potentially contribute ≥100% to the RNI of vitamin B_12_, EPA and DHA, and selenium, in addition to being able to contribute ≥50% to the RNIs of three or more nutrients simultaneously. All sampled species, aside from the demersal species, were also identified to potentially contribute ≥25% to the RNIs of six or more nutrients. This illustrates the wide diversity of nutrients that are available in marine fish and strengthens the evidence that marine fish provide high quantities of multiple critical nutrients required to improve micronutrient deficiencies. Further, fish enhances the bioavailability of minerals like iron and zinc from diets centred on starchy staples, thus including even small amounts of fish in the diet may enhance overall micronutrient bioavailability in the diet [74,76]. Due to the variability in micronutrient content with species, the nutritional quality needs to be examined for more marine fish species in order to optimise the complementary role fish may play in improving food and nutrition security in Bangladesh. 

### 4.5. Strengths and Limitations

Representable sampling and proficient nutrient analyses are fundamental to yield high quality FCD that may be incorporated into local FCT. In this paper, the analytical value of each nutrient is based on three pooled samples consisting of a total of between 15 and 840 individual fish depending on the biomass of the species. This number is well above the standard value of ten units that is generally viewed as an appropriate number of samples used for FCD. Additionally, the species fringescale sardinella, slender rainbow sardine, and Bombay duck were sampled from multiple locations. This provides more data for each species’ mean value, thus strengthening the confidence in the values [14]. However, pooled samples do not illustrate the individual variation within the species [77], and the nutrient content may also vary between different seasons, areas, maturity stages, and the sex of the fish [68,70]. All data presented in this paper were analysed at a national reference laboratory using accredited methods, thus representing important contributions to the Bangladeshi FCT and providing insight into the scarcely investigated nutrient composition of mesopelagic fish species. 

## 5. Conclusions

The nutrient composition of several macro and micronutrients in seven marine fish species from Bangladesh, which are not included in the FCT for Bangladesh, are described in this paper. Among these seven species, five are commonly consumed by the Bangladeshi people and two are mesopelagic species not yet evaluated for their potential contribution to local food and nutrition security. The two mesopelagic species had the highest content of all fatty acids including EPA and DHA, vitamin A, calcium, iron, iodine, and selenium, whereas the demersal species had the lowest content of total fat and crude protein, all fatty acids, and all vitamins and minerals of the included species All species, with the exception of the demersal species Bombay duck (9% dry matter), were identified to potentially contribute substantially to the RNI of women of reproductive age for multiple micronutrients simultaneously if included in the diet. Our findings also suggest that the diversity of micronutrients in marine fish varies widely by species. It is important to recognise that our results are reported for raw fish, and both the processing and preparation of meals may impact the nutrient content of the species. More studies on the nutritional composition of fish following the value chain from fresh to prepared meals are needed to better understand the potential contribution of commonly consumed marine fish species and new potential marine resources to contribute to local food and nutrition security in Bangladesh. 

## Figures and Tables

**Figure 1 foods-09-00730-f001:**
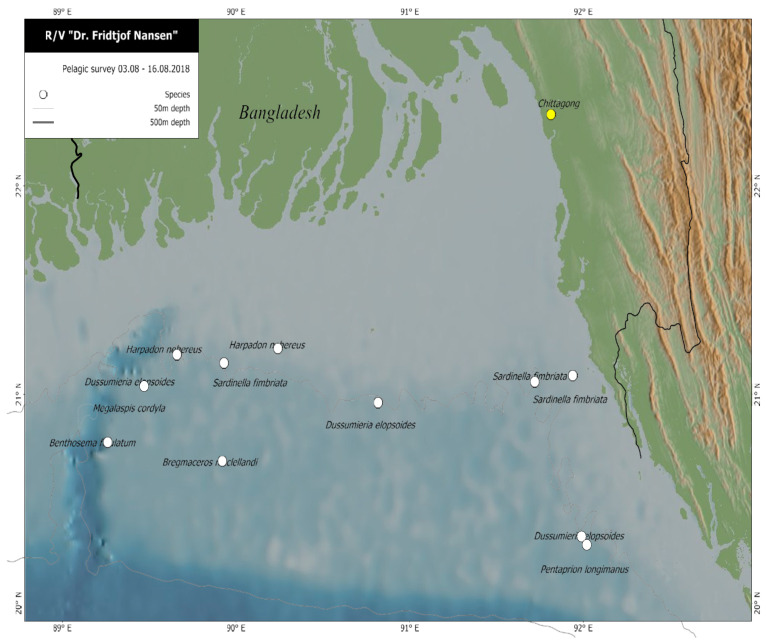
Sampling locations of the included fish species during the 2018 Nansen survey in Bangladesh, the Bay of Bengal.

**Figure 2 foods-09-00730-f002:**
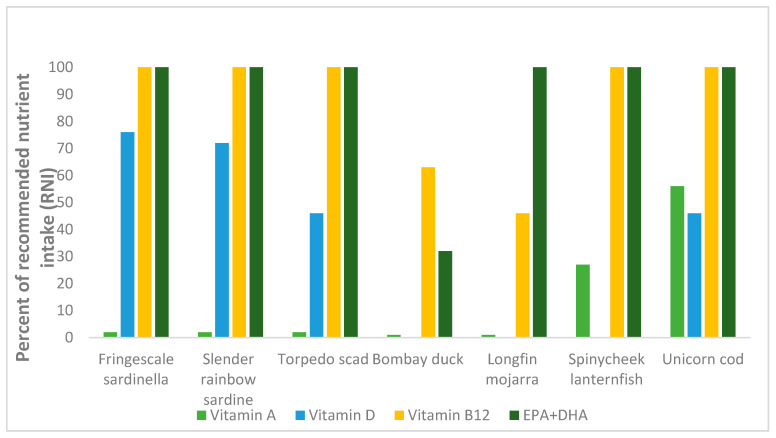
Percentage of recommended nutrient intake (RNI) for women of EPA + DHA and selected vitamins of 100 g raw, edible parts of Fringescale sardinella (fillet), Slender rainbow sardine (fillet), Torpedo scad (fillet), Bombay duck (fillet), Longfin mojarra (fillet), Spinycheek lanternfish (whole fish), and Unicorn cod (whole fish).

**Figure 3 foods-09-00730-f003:**
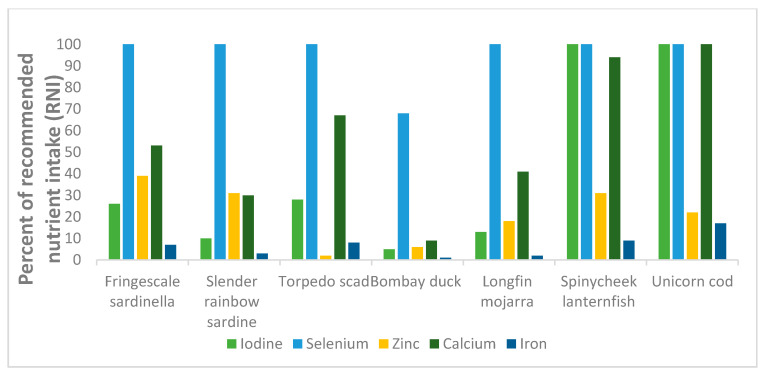
Percentage of recommended nutrient intake (RNI) for women of selected minerals and trace element for 100 g raw, edible parts of Fringescale sardinella (fillet), Slender rainbow sardine (fillet), Torpedo scad (fillet), Bombay duck (fillet), Longfin mojarra (fillet), Spinycheek lanternfish (whole fish), and Unicorn cod (whole fish).

**Table 1 foods-09-00730-t001:** Identification details and overview of fish species sampled from Bangladesh.

Common Name	Scientific Name	Local Name	Habitat	Fish Tissue Sampled	Number of Pooled Samples	Number of Fish in Each Pooled Sample
**Fringescale sardinella ^1^**	*Sardinella fimbriata*	Chapila	Pelagic	Fillet with skin and bones	3	23 ^a^
**Fringescale sardinella ^2^**	*Sardinella fimbriata*	Chapila	Pelagic	Fillet with skin and bones	3	25
**Fringescale sardinella ^3^**	*Sardinella fimbriata*	Chapila	Pelagic	Fillet with skin and bones	3	25
**Slender rainbow sardine ^1^**	*Dussumieria elopsoides*	Maricha	Pelagic	Fillet with skin and bones	3	25
**Slender rainbow sardine ^2^**	*Dussumieria elopsoides*	Maricha	Pelagic	Fillet with skin and bones	3	25
**Slender rainbow sardine ^3^**	*Dussumieria elopsoides*	Maricha	Pelagic	Fillet with skin and bones	3	23 ^a^
**Torpedo scad**	*Megalaspis cordyla*	Kauwa	Pelagic	Fillet with skin and bones	3	5
**Spinycheek lanternfish**	*Benthosema fibulatum*	Puiya	Mesopelagic	Whole fish	3	250
**Unicorn cod**	*Bregmaceros mcclellandi*	-^b^	Mesopelagic	Whole fish	3	280
**Bombay duck ^1^**	*Harpadon nehereus*	Loittya	Demersal	Fillet with skin and bones	3	25
**Bombay duck ^2^**	*Harpadon nehereus*	Loittya	Demersal	Fillet with skin and bones	3	20
**Longfin mojarra**	*Pentaprion longimanus*	Dom Mach	Demersal	Fillet with skin and bones	3	25

^a^ One pooled sample consisted of 22 fish, whereas the other two consisted of 23 fish. ^b^ Local name of fish species not available. ^1,2,3^ Same species sampled multiple times from different locations.

**Table 2 foods-09-00730-t002:** Mean physical parameters of fish species sampled from Bangladesh.

Common name	Weight (g) ^a^	Length (cm) ^a^
**Pelagic**		
Fringescale sardinella ^1^	35.2	16.1
Fringescale sardinella ^2^	40.3	16.4
Fringescale sardinella ^3^	43.5	16.4
Slender rainbow sardine ^1^	49.7	17.0
Slender rainbow sardine ^2^	72.6	20.3
Slender rainbow sardine ^3^	67.8	19.3
Torpedo scad	114.6	25.2
**Mesopelagic**		
Spinycheek lanternfish	0.6	<5
Unicorn cod	0.5	<6
**Demersal**		
Bombay duck ^1^	110.3	25.5
Bombay duck ^2^	117.7	24.2
Longfin mojarra	20.8	11.4

^a^ Weight and length measurements are expressed as the mean of one pooled sample consisting of n number of fish. ^1,2,3^ Same species sampled multiple times from different locations.

**Table 3 foods-09-00730-t003:** Analytical values of the proximate composition of the fish species sampled from Bangladesh ^a^.

Sampled Species		Dry Matter	Protein	Fat
**Pelagic**	n ^b^	%	g/100 g	g/100 g
Fringescale sardinella^1^	3	23.6 ± 0.5	21 ± 0.0	1.6 ± 0.2
Fringescale sardinella^2^	3	25.0 ± 0.3	21 ± 0.6	1.8 ± 0.1
Fringescale sardinella^3^	3	25.3 ± 0.4	21 ± 0.6	2.4 ± 0.4
**Mean of all Fringescale sardinella ^1,2,3^**	9	24.6 ± 0.9	21 ± 0.5	1.9 ± 0.4
Slender rainbow sardine ^1^	3	23.4 ± 2.2	21 ± 2.1	1.13 ± 0.1
Slender rainbow sardine ^2^	3	23.0 ± 0.3	21 ± 0.6	1.3 ± 0.3
Slender rainbow sardine ^3^	3	24.1 ± 0.4	21 ± 0.6	1.9 ± 0.1
**Mean of all Slender rainbow sardine ^1,2,3^**	9	23.5 ± 1.3	21 ± 1.1	1.4 ± 0.4
Torpedo scad	3	24.6 ± 1.8	21 ± 1.5	1.3 ± 0.2
**Mesopelagic**				
Spinycheek lanternfish	3	24.2 ± 0.5	17 ± 0.1	3.3 ± 0.1
Unicorn cod	3	20.9 ± 1.1	15 ± 1.1	1.6 ± 0.2
**Demersal**				
Bombay duck ^1^	3	8.12 ± 0.2	10 ± 1.8	0.64 ± 0.0
Bombay duck ^2^	3	9.45 ± 0.9	11 ± 2.4	0.85 ± 0.1
**Mean of all Bombay duck ^1,2^**	6	8.79 ± 0.9	10 ± 2.0	0.74 ± 0.1
Longfin mojarra	3	23.3 ± 1.0	19 ± 1.2	2.5 ± 0.1

^a^ Values are presented as means ± standard deviations (SD) of the fish species analysed in triplicates, expressed as the nutrient content per 100 g raw, edible part. ^b^ Number of pooled samples analysed. Each pooled sample consisted of a minimum of 5 fish. ^1,2,3^ Same species sampled multiple times from different locations.

**Table 4 foods-09-00730-t004:** Analytical values of the content of minerals and trace elements in the species sampled from Bangladesh ^a^.

Sampled Species		Ca	Na	K	Mg	P	I	Se	Zn	Fe
	n ^b^	mg/100 g	mg/100 g	mg/100 g	mg/100 g	mg/100 g	µg/100 g	µg/100 g	mg/100 g	mg/100 g
**Pelagic**										
Fringescale sardinella ^1^	3	597 ± 6	80 ± 2	470 ± 10	43 ± 1	583 ± 15	23.7 ± 0.6	113.3 ± 5.8	1.9 ± 0.2	2.2 ± 0.1
Fringescale sardinella ^2^	3	473 ± 117	84 ± 1	500 ± 10	44 ± 2	543 ± 51	66.7 ± 15.3	116.7 ± 5.8	2.0 ± 0.1	2.4 ± 0.2
Fringescale sardinella ^3^	3	527 ± 244	73 ± 2	487 ± 12	43 ± 3	557 ± 117	25.3 ± 0.6	99 ± 9.3	1.9 ± 0.2	1.9 ± 0.2
**Mean of all Fringescale sardinella ^1,2.3^**	9	532 ± 146	79 ± 5	486 ± 16	43 ± 2	561 ± 67	38.6 ± 22.4	110 ± 10	1.9 ± 0.2	2.2 ± 0.3
Slender rainbow sardine ^1^	3	353 ± 90	76 ± 7	503 ± 49	46 ± 4	473 ± 67	16.3 ± 0.6	84 ± 11	1.5 ± 0.2	1.1 ± 0.2
Slender rainbow sardine ^2^	3	280 ± 56	62 ± 3	500 ± 20	43 ± 2	430 ± 26	11.0 ± 0.0	70 ± 3.8	1.4 ± 0.1	1.0 ± 0.1
Slender rainbow sardine ^3^	3	277 ± 31	83 ± 6	513 ± 15	46 ± 1	443 ± 15	16.7 ± 0.6	75 ± 2.0	1.5 ± 0.1	0.9 ± 0.0
**Mean of all Slender rainbow sardine ^1,2,3^**	9	303 ± 67	74 ± 10	506 ± 28	45 ± 3	449 ± 41	14.7 ± 2.8	76 ± 8.4	1.5 ± 0.1	1.0 ± 0.1
Torpedo scad	3	670 ± 549	70 ± 17	417 ± 21	37 ± 8	580 ± 282	41.7 ± 16.3	63 ± 0.0	0.1 ± 0.3	2.4 ± 0.1
**Mesopelagic**										
Spinycheek lanternfish	3	940 ± 69	230 ± 0	303 ± 6	60 ± 1	577 ± 12	160.0 ± 26.5	120.0 ± 0.0	1.5 ± 0.1	2.5 ± 0.1
Unicorn cod	3	1033 ± 58	260 ± 10	327 ± 15	57 ± 2	613 ± 12	160.0 ± 20.0	67 ± 3.5	1.1 ± 0.0	4.9 ± 5.3
**Demersal**										
Bombay duck ^1^	3	126 ± 139	150 ± 10	177 ± 21	18 ± 2	143 ± 59	9.1 ± 0.6	15 ± 1.5	0.3 ± 0.1	0.2 ± 0.1
Bombay duck ^2^	3	46 ± 45	297 ± 50	210 ± 26	20 ± 1	126 ± 31	6.7 ± 0.7	18 ± 4.0	0.3 ± 0.0	0.2 ± 0.0
**Mean of all Bombay duck ^1,2^**	6	86 ± 102	173 ± 41	193 ± 28	19 ± 2	135 ± 43	7.9 ± 1.4	17 ± 3.0	0.3 ± 0.1	0.2 ± 0.1
Longfin mojarra	3	407 ± 93	78 ± 6	410 ± 30	38 ± 1	437 ± 32	19.0 ± 4.4	44 ± 2.0	0.9 ± 0.1	0.6 ± 0.0

^a^ Values are presented as means ± SD of the fish species analysed in triplicates, expressed as the nutrient content per 100 g raw, edible part. ^b^ Number of pooled samples analysed. Each pooled sample consisted of a minimum of 5 fish. ^1,2,3^ Same species sampled multiple times from different locations. **Abbreviations**: Ca: calcium, Fe: iron, I: iodine, K: potassium, Mg: magnesium, Na: sodium, P: phosphorus, SD: standard deviation, Se: selenium, Zn: zinc.

**Table 5 foods-09-00730-t005:** Analytical values of the vitamin A (A_1_ + A_2_), vitamin B_12_, and vitamin D content in species sampled from Bangladesh ^a^.

Species ^a^		Vitamin A_1_	Vitamin A_2_	Vitamin B_12_	Vitamin D
	n ^b^	µg/100 g	µg/100 g	µg/100 g	µg/100 g
**Pelagic**					
Fringescale sardinella ^1^	3	10.3 ± 0.6	5.6 ± 2.1	4.1 ± 0.3	4.7 ± 0.6
Fringescale sardinella ^2^	3	12.7 ± 4.7	8.7 ± 1.5	11 ± 3.3	4.0 ± 1.0
Fringescale sardinella ^3^	3	14.3 ± 1.5	11.0 ± 2.0	14 ± 2.1	2.7 ± 0.6
**Mean of all Fringescale sardinella ^1,2,3^**	9	12.4 ± 3.0	8.4 ± 2.9	9.9 ± 4.9	3.8 ± 1.1
Slender rainbow sardine ^1^	3	7.3 ± 1.5	1.8 ± 0.2	8.3 ± 0.4	3.3 ± 0.6
Slender rainbow sardine ^2^	3	4.1 ± 2.4	1.3 ± 0.5	7.1 ± 1.4	2.7 ± 1.2
Slender rainbow sardine ^3^	3	16.3 ± 1.5	1.4 ± 0.3	8.8 ± 0.5	4.7 ± 2.1
**Mean of all Slender rainbow sardine ^1,2,3^**	9	9.3 ± 5.7	1.5 ± 0.4	8.1 ± 1.1	3.6 ± 1.5
Torpedo scad	3	11.7 ± 4.0	0.7 ± 0.2	15 ± 2.1	2.3 ± 0.6
**Mesopelagic**					
Spinycheek lanternfish	3	133 ± 49.3	6.2 ± 2.4	13 ± 1.5	<1.0 ^d^
Unicorn cod	3	280 ± 26.5	8.7 ± 1.2	7.9 ± 0.7	2.3 ± 0.6
**Demersal**					
Bombay duck ^1^	3	5.2 ± 0.8	<0.5 ^d^	1.2 ± 0.1	<1.0 ^d^
Bombay duck ^2^	3	9.3 ± 2.9	<0.5 ^d^	1.7 ± 0.6	<1.0 ^d^
**Mean of all Bombay duck ^1,2^**	6	7.3 ± 3.0	<0.5 ^d^	1.5 ± 0.5	<1.0 ^d^
Longfin mojarra	3	7.3 ± 2.6	<0.5 ^d^	1.1 ± 0.2	<1.0 ^d^

^a^ Values are presented as means ± SD of the 19 fish species analysed in triplicates, expressed as the nutrient content per 100 g raw, edible part. ^b^ Number of pooled samples analysed. Each pooled sample consisted of a minimum of 5 fish. ^d^ Value below LOQ (Vitamin A: 0.5 µg/100 g raw, edible parts; Vitamin D: 1.0 µg/100 g raw, edible parts). ^1,2,3^ Same species sampled multiple times from different locations.

**Table 6 foods-09-00730-t006:** Analytical values of the fatty acid composition of fish species sampled from Bangladesh ^a^.

Sampled Species		Sum SFA	Sum MUFA	Sum PUFA	Sum n-3	Sum n-6	EPA	DHA
	n ^b^	g/100 g (% ^c^)	g/100 g (% ^c^)	g/100 g (% ^c^)	g/100 g (% ^c^)	g/100 g (% ^c^)	g/100 g (% ^c^)	g/100 g (% ^c^)
**Pelagic**								
Fringescale sardinella ^1^	3	0.44 ± 0.06 (39)	0.16 ± 0.06 (14)	0.47 ± 0.05 (42)	0.37 ± 0.03 (34)	0.09 ± 0.02 (7.3)	0.07 ± 0.01 (5.8)	0.27 ± 0.01 (24.2)
Fringescale sardinella ^2^	3	0.49 ± 0.03 (42)	0.18 ± 0.03 (15)	0.46 ± 0.02 (39)	0.36 ± 0.01 (31)	0.08 ± 0.01 (7.3)	0.08 ± 0.01 (6.7)	0.25 ± 0.03 (21)
Fringescale sardinella ^3^	3	0.88 ± 0.14 (40)	0.39 ± 0.10 (18)	0.80 ± 0.10 (37)	0.64 ± 0.08 (29)	0.14 ± 0.01 (6.3)	0.20 ± 0.05 (8.9)	0.34 ± 0.02 (16)
**Mean of all Fringescale sardinella ^1,2,3^**	9	0.60 ± 0.23 (40)	0.24 ± 0.13 (16)	0.57 ± 0.18 (39)	0.46 ± 0.14 (32)	0.10 ± 0.03 (7.1)	0.11 ± 0.07 (7.1)	0.29 ± 0.05 (20)
Slender rainbow sardine ^1^	3	0.37 ± 0.02 (40)	0.11 ± 0.00 (12)	0.44 ± 0.03 (46)	0.37 ± 0.02 (39)	0.07 ± 0.00 (7.4)	0.06 ± 0.00 (6.5)	0.28 ± 0.02 (30)
Slender rainbow sardine ^2^	3	0.36 ± 0.03 (40)	0.11 ± 0.01 (12)	0.40 ± 0.04 (46)	0.34 ± 0.03 (39)	0.06 ± 0.01 (7.0)	0.06 ± 0.01 (6.7)	0.26 ± 0.02 (29)
Slender rainbow sardine ^3^	3	0.57 ± 0.03 (40)	0.17 ± 0.01 (12)	0.62 ± 0.06 (44)	0.52 ± 0.05 (37)	0.10 ± 0.01 (7.0)	0.10 ± 0.01 (7.2)	0.38 ± 0.04 (27)
**Mean of all Slender rainbow sardine ^1,2,3^**	9	0.43 ± 0.10 (40)	0.13 ± 0.03 (12)	0.49 ± 0.11 (45)	0.41 ± 0.09 (38)	0.08 ± 0.02 (7.1)	0.07 ± 0.02 (6.8)	0.31 ± 0.06 (29)
Torpedo scad	3	0.35 ± 0.05 (34)	0.14 ± 0.02 (14)	0.48 ± 0.03 (47)	0.38 ± 0.02 (37)	0.10 ±0.01 (10)	0.05 ± 0.00 (5.1)	0.28 ± 0.01 (27)
**Mesopelagic**								
Spinycheek lanternfish	3	0.88 ± 0.00 (40)	0.41 ± 0.00 (19)	0.85 ± 0.01 (39)	0.68 ± 0.01 (31)	0.17 ± 0.00 (7.8)	0.16 ± 0.00 (7.2)	0.43 ± 0.01 (19)
Unicorn cod	3	0.31 ± 0.02 (31)	0.16 ± 0.01 (16)	0.45 ± 0.02 (45)	0.37 ± 0.02 (37)	0.08 ± 0.00 (7.8)	0.06 ± 0.00 (6.1)	0.28 ± 0.01 (28)
**Demersal**								
Bombay duck ^1^	3	0.11 ± 0.02 (39)	0.06 ± 0.01 (19)	0.10 ± 0.01 (36)	0.074 ± 0.01 (26)	0.03 ± 0.00 (9.3)	0.02 ± 0.00 (5.8)	0.05 ± 0.00 (17)
Bombay duck ^2^	3	0.21 ± 0.07 (43)	0.11 ± 0.04 (22)	0.13 ± 0.01 (29)	0.10 ± 0.01 (22)	0.03 ± 0.00 (7.1)	0.03 ± 0.01 (5.6)	0.06 ± 0.01 (14)
**Mean of all Bombay duck ^1,2^**	6	0.16 ± 0.07 (41)	0.08 ± 0.04 (21)	0.12 ± 0.02 (32)	0.09 ± 0.02 (24)	0.03 ± 0.01 (8.2)	0.02 ± 0.01 (5.7)	0.06 ± 0.01 (16)
Longfin mojarra	3	0.74 ± 0.05 (38)	0.45 ± 0.4 (23)	0.67 ± 0.03 (34)	0.52 ± 0.02 (27)	0.15 ± 0.00 (7.6)	0.08 ± 0.00 (4.2)	0.38 ± 0.02 (20)

^a^ Values are presented as means ± SD of the fish species analysed in triplicates, expressed as the nutrient content per 100 g raw, edible parts. ^b^ Number of pooled samples analysed. Each pooled sample consisted of a minimum of 5 fish. ^c^ Values are given in percent of total fatty acids. ^1,2,3^ Same species sampled multiple times from different locations. Abbreviations: DHA: docosahexaenoic acid; EPA: eicosapentaenoic acid; MUFA: monounsaturated fatty acids; PUFA: polyunsaturated fatty acids; SD: standard deviation, SFA: saturated fatty acids.

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
