# Peer review of "Nutrient Composition of Demersal, Pelagic, and Mesopelagic Fish Species Sampled Off the Coast of Bangladesh and Their Potential Contribution to Food and Nutrition Security—The EAF-Nansen Programme"

_foods, 2020, doi:10.3390/foods9060730_

Round 1

Reviewer 1 Report

Dear authors,

This manuscript is very well written and presented and provides invaluable information which could potentially contribute to improving Food and Nutrition security in Bangladesh and elsewhere. The analysis of mesopelagic fish was novel and of considerable interest. The micronutrient content of the mesopelagic fish was impressive and very important to publish in my opinion.

I have made several comments in the PDF attached. One comment was that I was wondering if the preparation of the fish might affect the macro and micro nutrient concentrations? i.e., the mesopelagic fish were analysed whole whereas the other fish were filleted?

Can the mesopelagic fish be obtained easily when trawling and, if so, can sufficient quantities be obtained when fishing, given that they are so small, to make them a viable option for either human nutrition or aquaculture feed?

In the analytical methods section, where a method doesn't have an associated reference you need to include more detail - what column was used for HPLC analysis?, what were the mobile phases used?, temperature?, gradient conditions? detector wavelength? The reader should be able to reproduce the methods relatively easily. This issue occurs in several places in this section.

Author Response

Dear reviewers and Editor,

Thank you for reviewing our manuscript. We have revised the manuscript according to the reviewer’s suggestions. Additionally, we have corrected a few words written in American English to be written in British English and inserted two more references (references 63 and 64).

Comments from Reviewer 1

Point 1: This manuscript is very well written and presented and provides invaluable information which could potentially contribute to improving Food and Nutrition security in Bangladesh and elsewhere. The analysis of mesopelagic fish was novel and of considerable interest. The micronutrient content of the mesopelagic fish was impressive and very important to publish in my opinion.

I have made several comments in the PDF attached. One comment was that I was wondering if the preparation of the fish might affect the macro and micro nutrient concentrations? i.e., the mesopelagic fish were analysed whole whereas the other fish were filleted?

Response 1: Thank you very much. Yes, this is something we believe as well. This has also been mentioned in the discussion for mesopelagic fish species (line 336-340). We have in a previous study evaluated the difference between fish species that are commonly consumed whole and species where only the fish fillet is consumed and have discovered significant differences in nutrient density. We have now inserted this reference (reference 64) in line 340.

Point 2: Can the mesopelagic fish be obtained easily when trawling and, if so, can sufficient quantities be obtained when fishing, given that they are so small, to make them a viable option for either human nutrition or aquaculture feed?

Response 2: The biomass of mesopelagic fish has recently been estimated to be equivalent to 100 times the annual catch of traditional fisheries. Most mesopelagic species remain fairly scattered in the water column, making it difficult to catch large quantities in one trawl, whereas new data suggest that some species may be present in large quantities at the same place in the water column. The data on biomass estimation and capture technologies of mesopelagic resources are very novel (as are the data on the chemical composition of mesopelagic species) and needs to be further developed in order to make the harvesting of mesopelagic resources efficient and sustainable. A lot of research is currently ongoing for establishing more knowledge on mesopelagic resources and their potential as future food and feed resources. The research is currently on a very preliminary level, but the results look promising.

Point 3: In the analytical methods section, where a method doesn't have an associated reference you need to include more detail - what column was used for HPLC analysis?, what were the mobile phases used?, temperature?, gradient conditions? detector wavelength? The reader should be able to reproduce the methods relatively easily. This issue occurs in several places in this section.

Response 3: Thank you for commenting on this. As referenced to in the second sentence of the analytical methods section, a methodology paper describing all analytical methodology and protocols used for nutrient and contaminant analyses under the EAF-Nansen Programme is currently under review in MethodsX. In this paper, these parameters are described in detail, so readers are able to reproduce the analyses. The aim of this methodology paper is to assemble all the protocols and descriptions of the methodological procedures in one paper to avoid the need to describe the same methodology at great lengths repeatedly for every paper published under the EAF-Nansen Programme. This methodology paper will allow us to significantly shorten the methodology text (as we have in this paper from Bangladesh), as we have experienced with previously published papers are often requested by reviewers. If requested by the reviewer, we can make this methodology paper available for the reviewer to read.

Point 4 (PDF): Please put space inbetween Themesopelagic - i.e., The mesopelagic

Response 4: Thank you - corrected.

Point 5 (PDF): Will preparing these mesopelagic fish whole, which includes the head, skin, tail and viscera, affect the content of macro and micro nutrients? I appreciate the fish are very small to fillet and dissect.

Response 5: Please see response 1.

Point 6 (PDF): include location of IMR for clarity again

Response 6: “Bergen, Norway” has been inserted in the sentence for clarity.

Point 7 (PDF): Is protein measured using a protein analyser with standardised methodology? e.g., we use the Kjeldahl method with Foss equipment in our labs

Response 7: Thank you for this question. Yes, protein is measured with standardised methodology at the IMR laboratories. As mentioned in the methodology paper for the EAF-Nansen Programme that is currently under review, all methods performed are accredited to ISO 17025:2005, and the laboratories regularly participate in national and international proficiency tests to assess the accuracy and precision of the analyses. However, we do not use the Kjeldahl method; we use a combustion method for the determination of crude protein. This methodology is recommended by AOAC International, and a comparison of the two methods at 12 different laboratories have been performed (https://europepmc.org/article/med/8374325, reference #31 in the manuscript file), where all laboratories and instruments showed satisfactory results.

Point 8 (PDF): Where a method isn't referenced the authors need to include the column used, the mobile phase(s), temperature - the reader should be able to reproduce the analysis relatively easily. This issue occurs throughout the material and methods from this point onwards.

Response 8: Please see response 3.  

Point 9 (PDF): It would be very interesting to determine how these fish contribute to RNI's of children. Its not necessary to include for this manuscript - just my observations

Response 9: We very much agree.

Point 10 (PDF): I would change the words 'highly needed' to 'crucial', i.e., ..these types of studies are crucial

Response 10: The wording has been changed from “highly needed” to “crucial”.

Comments from Reviewer 2

Point 1: This study shows how local marine fish can contribute essential nutrients to the diet of women of reproductive age in Bangladesh. This contributes information that can help improve the intake of nutrients in a population at risk of malnutrition.

The strengths of the study are good design and methods, and discussion of results. Data are presented well.

One improvement could be more statisical analysis of the results, such as ANOVA for comparison of values between fish species.

Response 1: Thank you very much. With this study, we hope to promote the consumption of fish as an important and highly nutrient-dense source of vital micronutrients by documenting the nutrient composition of fish and including high-quality analytical data in local food composition tables. The goal of this study was to document the nutrient content of various commonly consumed fish species to promote the consumption of the food group as a whole, not to document which fish species are superior to other fish species in terms of nutrient content. Furthermore, as the fish sampled for this paper were prepared and analysed as composite samples and not individual samples, some of the biodiversity and individual variation within a species will be masked when performing statistical analyses to compare nutrient density. We also have a low n (three composite samples per species), so drawing conclusions from significant differences between the species may be inaccurate. Therefore, we believe not including any statistical analyses would be preferred for this paper.

Reviewer 2 Report

This study shows how local marine fish can contribute essential nutrients to the diet of women of reproductive age in Bangladesh. This contributes information that can help improve the intake of nutrients in a population at risk of malnutrition. 

The strengths of the study are good design and methods, and discussion of results. Data are presented well. 

One improvement could be more statisical analysis of the results, such as ANOVA for comparison of values between fish species. 

Author Response

see attachments
